# How Low Can LoRA Go: System-Level Throughput, Energy, and Model Quality Tradeoffs when Fine-Tuning Adapters

Connor Espenshade*, Umesh Deshpande†, Yue Zhu†, Eun Kyung Lee†, Martha A. Kim*

* Columbia University. *connor.espenshade@columbia.edu, martha@cs.columbia.edu*
† IBM Research. *udeshpa@ibm.com, yue.zhu@ibm.com, eunkyung.lee@us.ibm.com*

*Abstract*—As models scale beyond trillions of parameters, extending their functionality is increasingly achieved through fine-tuning existing base models. However, fine-tuning all parameters remains computationally expensive. Recent techniques such as Low-Rank Adaptation (LoRA) have been developed to reduce the number of trainable parameters. LoRA adapters have gained widespread adoption, but their effects on GPU system metrics, such as throughput and energy efficiency, are not yet well understood.

In this study, we examine these system-level metrics as a function of the LoRA adapter rank. Our findings show that reducing the rank of LoRA adapters does not lead to a significant drop in model quality, while simultaneously improving throughput, energy efficiency, and memory usage by up to 2.7x. Further, we find that the presence of a LoRA adapter, rather than its rank size, can greatly improve model quality compared to a zero-shot inference base model. This makes smaller LoRA adapters a compelling choice from both a system and a model quality perspective.

## I. INTRODUCTION

As large language models (LLMs) continue to grow in size, the time and computational resources required to train them from scratch have increased proportionally. For many general tasks, these models already possess sufficient capabilities to meet performance requirements. However, for more specialized tasks, such as summarization, or domain-specific applications, such as legal or medical contexts, the model often requires additional adaptation. In these cases, the base model can be *fine-tuned* to extend its general knowledge with the specific requirements of the target task or domain [6].

Due to the substantial size of modern LLMs, fine-tuning all of its parameters remains computationally expensive. To address this, *parameter-efficient fine-tuning* (PEFT) techniques adapt the base model by training a small number of additional parameters, reducing compute and memory requirements. This not only makes fine-tuning more lightweight but also results in compact adapters that are easier to store and distribute.

This paper examines the GPU-level implications of fine-tuning and inference using Low-Rank Adaptation (LoRA) adapters. LoRA introduces trainable low-rank matrices into the model, effectively reducing the dimensionality of the adaptation layer, which can then be projected back to match the original parameter space. We analyze the impact of LoRA adapter rank on both fine-tuning and inference performance

across two benchmark tasks: general language modeling to reproduce Wikipedia text from the Wikitext dataset and extractive question answering to answer questions based on given context from the SQuAD dataset [11], [12].

While prior work has studied the effectiveness of LoRA in terms of model quality and architecture design [4], [7], its influence on systems-level behavior—such as GPU compute utilization, latency, and memory consumption—remains less well understood. This paper aims to bridge that gap by quantifying how these performance metrics, along with model quality, vary as a function of the LoRA adapter rank.

In this study, we find that system performance does not significantly vary even when fine-tuning or running inference on models thousands of times larger. As we sweep ranks from 1 to 2048, we find that increasing ranks early on can improve model quality and accuracy without negatively impacting throughput, memory, or energy. However, when larger ranks are reached, the returns from increased model quality diminish or even reverse. In fine-tuning, we find that variance occurs largely when batch size changes, suggesting strong links between parallelism and LoRA capability.

## II. BACKGROUND ON LOW-RANK ADAPTATION (LoRA)

Low-Rank Adaptation (LoRA) is a parameter-efficient fine-tuning technique that reduces the number of trainable parameters by approximating updates to weight matrices using low-rank decompositions.

In standard fine-tuning, the output of a pre-trained model is given by $h = W_0 x$, where $W_0$ is a fixed, pre-trained weight matrix and $x$ is the input [7]. Fine-tuning introduces a learnable adjustment $\Delta W$ to the base model weights, yielding an updated output:

$$h' = (W_0 + \Delta W)x = W_0 x + \Delta W x.$$

In LoRA, the adjustment $\Delta W$ is not learned directly. Instead, it is expressed as the product of two lower-dimensional matrices, $A$ and $B$, such that $\Delta W = AB$. Given an original weight matrix $W_0 \in \mathbb{R}^{d \times k}$, LoRA defines $A \in \mathbb{R}^{d \times r}$ and $B \in \mathbb{R}^{r \times k}$, where $r \ll \min(d, k)$. This formulation reduces the number of trainable parameters from $d \times k$ (full-rank) to $2 \times r \times \max(d, k)$ in the worst case [7], significantly lowering memory and computational requirements.

For example, in the LLaMA 3.1-8B model, the query and output projection matrices in the attention modules have dimensions $d = k = 4096$ [5]. Using LoRA with a rank of $r = 16$, the number of trainable parameters per attention matrix drops from $d^2 = 16,777,216$ to $2 \times d \times r = 131,072$, representing a 99.2% reduction. This leads to significant savings in terms of optimization during fine-tuning, storage of adapter weights, and computational cost during inference.

## III. METHODOLOGY

In this section, we describe our experiment setup. All fine-tuning and inference workloads were run for one full epoch on an NVIDIA A100-80GB, with Intel(R) Xeon(R) Platinum 8260 CPU @ 2.4GHz.

### A. Base Model

We use LLaMA 3.1-8B as the base model, selected for its compatibility with the A100-80GB GPU as it fits within memory without requiring quantization. It uses a $d = 4096$ embedding dimension, 32 attention heads, and 8 key-value heads [5]. We use the base non-instruct version to isolate the effects of fine-tuning.

### B. Fine-Tuning Datasets

To evaluate the impact of LoRA adapters at different ranks, we fine-tuned two distinct datasets across a range of adapter rank values.

*1) WikiText-2:* WikiText-2 is a language modeling dataset of 720 curated Wikipedia articles, totaling approximately 2.5 million tokens [11]. In our setup, the fine-tuned model can be prompted with a phrase corresponding to a Wikipedia topic, and it responds by generating text as if writing a Wikipedia-style article on that subject. As such, adapters fine-tuned on WikiText-2 reflect general-purpose language modeling ability and serve as a strong baseline for evaluating adapter performance. High-quality adapters trained on this dataset tend to perform well on next-token prediction tasks in decoder-style architectures.

WikiText-2 models are evaluated based on the loss computed over test samples, or the natural log of this loss, which is defined as their perplexity [2], [3]. Perplexity measures the model's confidence in predicting the next token in a sequence, with lower perplexity indicating greater certainty and accuracy. As such, it serves as a strong indicator of the language modeling quality of an adapter trained on WikiText-2.

*2) SQuAD:* The Stanford Question Answering Dataset (SQuAD) is a benchmark dataset for extractive question answering (QA) tasks [12]. Each of the 98,200 samples consists of a context paragraph describing a story, event, or topic, a question, and an answer. The answer is extractive—that is, it appears as a span directly within the context text. To ensure consistency in training and inference performance across samples, the context and answer are either truncated or padded to a fixed length of 512 tokens, depending on whether the input is too long or too short for the model. This

normalization is standard across BERT and HuggingFace as GPU kernels and accelerators perform better on fixed-length inputs [9].

Adapters fine-tuned on SQuAD are evaluated using SQuAD's official evaluation script, which measures both exact match (EM) and F1 score. The exact match metric checks whether the model's predicted answer matches the ground truth span verbatim, while the F1 score accounts for partial overlap between the predicted and true answers [12]. This paper uses SQuAD v2.0, which includes unanswerable questions where no answer is present in the context. For simplicity and since our focus is on the impacts on system performance, our experiments focus exclusively on the subset of questions that have valid answers within the context.

### C. Evaluation Workloads

For each workload, we conducted fine-tuning and inference experiments using LoRA adapters for WikiText-2 and SQuAD with ranks ranging from 1 to 2048 by powers of two. A maximum rank of 2048 was chosen because it corresponds to the full reconstruction of all parameters initially present in the model. Specifically, if the matrices $A$ and $B$ have dimensions $d \times r = 4096 \times 2048$, the total number of weights stored in the two matrices ($2 \times d \times r$) equals the total number of weights $d^2 = 4096^2$ present during full fine-tuning. We establish this as a baseline, and report our measurements of smaller adapters normalized to $r = 2048$.

We initially attempted to use full-parameter fine-tuning without LoRA as the baseline. However, these workloads consistently encountered out-of-memory errors and could not fit on a single GPU without substantial optimizations beyond those applied in the other experiments. As a result, we excluded full-parameter fine-tuning from the comparison. Therefore, the baseline for our study is the LoRA adapter with the largest rank that mirrors the total number of parameters in the fully fine-tuned model, i.e., $r = 2048$.

### D. Hyperparameter Selection

LoRA was applied to the attention module's weight matrices, specifically, the query (`q_proj`), key (`k_proj`), value (`v_proj`), and output (`o_proj`) projections. No LoRA adapters were applied to the feed-forward network layers.

The learning rate for both workloads was set to $5 \times 10^{-5}$, consistent with prior work on parameter-efficient fine-tuning [10]. Batch size during fine-tuning was workload dependent (discussed in results).

LLaMA-based tokenizers padded all prompts to a fixed token length of 512 for both workloads to ensure consistency across training samples. All fine-tuning and inference used mixed precision during computations: compute steps were performed using 16-bit floating point (fp16) precision (half precision) to speed up computation, yet the inputs and outputs remained 32-bit floats [1]. We elected to save quantization for future investigations, as quantization often decreases accuracy as a trade-off for lower memory usage, where we wanted to focus on the impacts of LoRA ranks for general models [8].

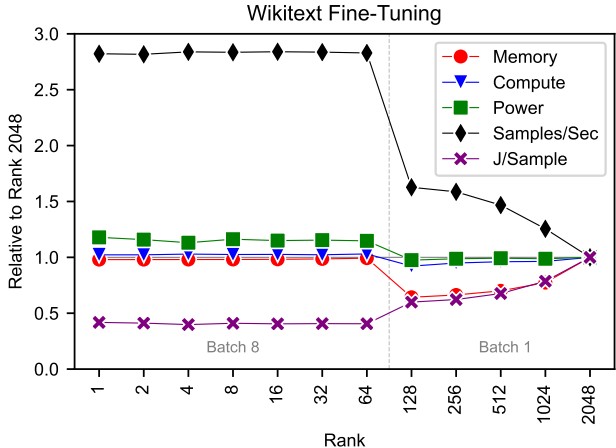

Fig. 1. Fine-tuning Wikitext-2 shows constant system performance for batch 8, a sharp degradation at the change in batch size, and an accelerating decline in system performance as ranks surpass 256.

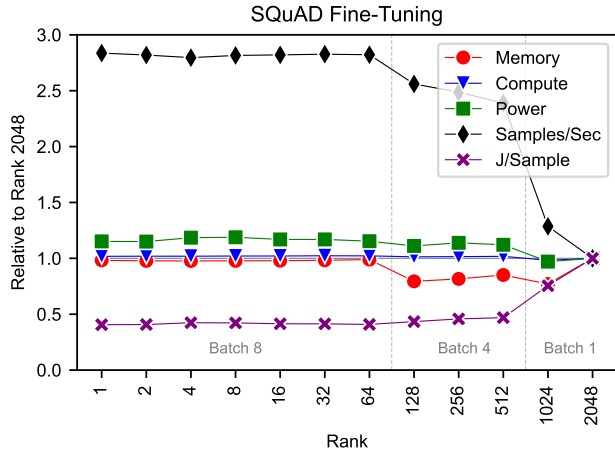

Fig. 2. Fine-tuning SQuAD across 3 batch sizes shows more consistency in system performance compared to Fig. 1. However, larger ranks necessitate smaller batch sizes, which still significantly impact throughput and energy.

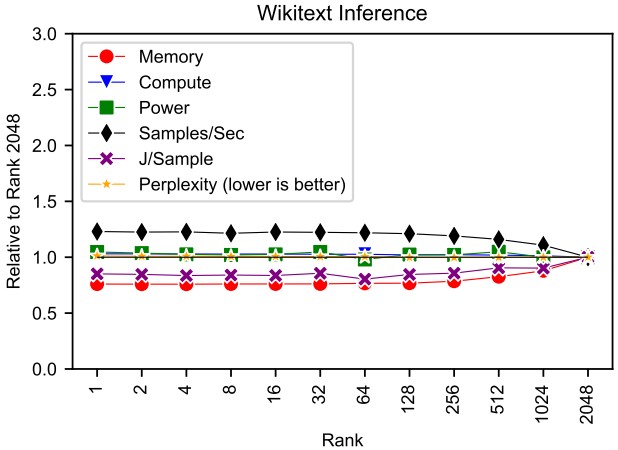

Fig. 3. Inference on Wikitext-2 demonstrates stark consistency across all ranks up to 512. Perplexity, measuring model quality where lower is better, remains unchanged from ranks 1 to 2048.

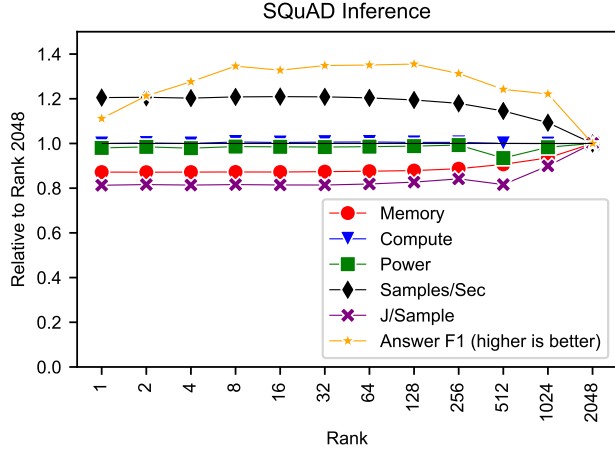

Fig. 4. Inference on SQuAD yields highest F1 score on rank 8, with system performance tradeoff for ranks less than 256. However, F1 quality declines after rank 128, showing small ranks remain optimal.

### E. Performance Metrics

In each experiment, we collected both GPU metrics and model accuracy/performance metrics. For GPU metrics, we use `nvidia-smi` to query GPU power, memory usage, and GPU utilization at four times per second for the duration of the workload. For model accuracy metrics, we tracked the model loss, perplexity and F1 score when using WikiText-2 and SQuAD, respectively. For model performance metrics, we gathered number of samples processed per second.

## IV. RESULTS & ANALYSIS

For all measurements, each workload is normalized to the largest adapter, with rank 2048. This corresponds to the full parameter count of the base model. All measurements are normalized to this value, with smaller ranks representing reductions in the number of parameters in the adapter.

### A. Fine-Tuning Performance Flat within a Given Batch Size When Varying Rank

Fine-tuning adapters on WikiText-2 and SQuAD, shown in Figures 1 and 2, respectively, results in minimal differences in system-level performance as adapter rank is increased, despite substantial changes in adapter size. Figure 1 demonstrates that for WikiText-2, reducing the rank from 64 to 1 decreases the adapter size by 64×, from 218 MB to 3.4 MB, without any noticeable change in throughput or energy per sample. This indicates that fine-tuning adapters with rank 1 or 64 yields equivalent training performance, making smaller adapters preferable. Similarly, for SQuAD (Figure 2), there is negligible impact on memory and compute usage, average power, and time and energy per sample. This suggests that fine-tuning 64× more parameters has no meaningful effect on overall system performance.

## B. Large Ranks Require Small Batches, Limiting Fine-Tuning Performance

During fine-tuning, the batch size was set to the maximum that could fit on a single NVIDIA A100 GPU (80 GB). For WikiText-2, a batch size of 8 was used for ranks $r < 100$, and a batch size of 1 for $r > 100$. For SQuAD, batch size had to be reduced more incrementally with increasing rank: 8 for $r < 100$, 4 for $100 < r < 1000$, and 1 for $r > 1000$. These batch size cutoffs are indicated by gray dashed lines in Figures 1 and 2.

We observe exceptions to the general at the highest ranks and smallest batch sizes. In WikiText-2, a sharp drop in performance is observed when the batch size decreases from 8 to 1. Specifically, throughput decreases by $1.5\times$ between ranks 64 and 128, whereas similar rank increases elsewhere produce no noticeable throughput change. At this inflection point, overall GPU memory usage also drops by approximately 30%, reducing efficiency and slightly increasing energy per sample. Beyond rank 128, both memory usage and energy per sample continue to increase following a parabolic trend, while throughput mirrors that pattern in its decline.

In the case of SQuAD, fine-tuning occurs across three distinct batch size regions, resulting in a more gradual decline in throughput. However, each reduction in batch size to accommodate the increasing number of parameters at higher ranks leads to a noticeable drop in relative throughput. For example, rank 2048 processes $3\times$ fewer samples per second than rank 1. Additionally, rank 2048, which uses the smallest batch size, shows a $2.1\times$ increase in relative energy per sample.

So, in terms of fine-tuning performance, batch size emerges as a key limiter for larger adapters. As rank increases, the resulting adapter size constrains parallelism during training, leading to significantly reduced throughput and increased energy consumption. From an energy standpoint, the GPU's energy per sample remains relatively stable until an inflection point at rank 512, beyond which energy consumption begins to rise more noticeably.

## C. Inference Performance Is Consistent Across Ranks

Inference results for WikiText, shown in Figure 3, are even more consistent, with minimal variance up to rank 256. All evaluations use a constant batch size of 4 to ensure comparability across ranks. In addition to performance, we assess model quality during inference. Perplexity improves slightly, decreasing from 8.04 at rank 1 to 7.92 at rank 128, a relative improvement of 1.5%, then stabilizes at this level for higher ranks.

Turning to SQuAD inference results in Figure 4, model quality, as measured by the Answer F1 score, increases for ranks 1 through 8, before remaining constant and even declining as system performance begins to worsen between ranks 128 to 2048. The F1 score improves by 21% from rank 1 to a peak at rank 8, representing a $1.4 times$ improvement over the full-parameter (rank=2048) version. From rank 8 to 64, the F1 score remains stable before dropping at higher ranks. Notably,

the rank 2048 adapter performs worse than even the smallest rank 1 model in terms of F1 score.

System performance for SQuAD inference follows a similar pattern, but with the starker effects appearing at higher ranks: initially negligible for smaller ranks, eventually noticeable for larger ranks. Energy per sample and throughput are both 20% better than the full parameter rank 2048 adapter for ranks 1 to 256. At rank 512, these values begin converging to weaker performance through rank 2048, with memory also rising 15% compared to all other adapters, with inference batch size is kept constant at 8.

## D. Adapter Presence Matters More than Size

Given our observations of negligible inference quality and performance changes for adapters with $256\times$ fewer bits, one explanation could be that the LLaMA-3.1-8B base model itself has enough information to generate Wikipedia-like summaries and answer SQuAD's questions. If that were the case, the base model alone should yield a similar perplexity and F1 score. However, for Wikitext-2, zero-shot inference has a perplexity $3.5\times$ higher than any Wikitext LoRA adapter and over $200\times$ greater than the difference in perplexity between rank 1 and 2048. Similarly for SQuAD, the base model has an F1 score 7.7% lower than the rank 1 adapter and 30% lower the peak F1 at rank 8. Here, moving from no adapter to a rank 1 adapter yields a larger bump in inference quality than increasing the adapter by $64\times$ from rank 8 to 256.

Consequently, we find that LoRA adapters are extremely efficient: very small adapters (i.e., rank 1 for Wikitext-2 and rank 8 for SQuAD) achieve strong model quality while providing the lowest energy consumption and highest throughput. Increasing the adapter size can improve model quality without impacting performance, up to $64\times$ the size of the original adapter. However, beyond this point, further increases yield either marginal improvements and potentially reductions in quality.

## V. FUTURE WORK

To more fully understand the impact of LoRA rank on system performance, more workloads could give an enhanced view of if some applications do have systems tradeoffs. Indeed, preliminary experiments of a summarization adapter fine-tuned on the CNN-DailyMail dataset have resulted in lower ROUGE scores compared to the zero-shot inference model, where we run inference against the "raw" base model.

## VI. CONCLUSION

In this study, we study the relation between system-level performance and model quality as we enlarge the internal dimension of LoRA adapters for fine-tuning. We find that there is little variance in model quality and system performance, but that including a LoRA adapter significantly improves model quality.

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
