# OpenReview forum: "How Low Can LoRA Go: System-Level Throughput, Energy, and Model Quality Tradeoffs when Fine-Tuning Adapters"
_iscaconf.org/ISCA/2025/Workshop/MLArchSys — MLArchSys 2025 Oral_

### Official Review · Reviewer_xumC · 2025-05-18
**This paper takes a first step towards understanding GPU-level, system-level, and energy implications of LoRA fine tuning, but does not contextualize previous work or clearly distill key findings/takeaways.**

**Confidence:** 4
**Rating:** 4

**Detailed Feedback And Questions For Authors:**

Thank you for submitting your work to MLArchSys'25! I enjoyed reading this paper. Below are questions and feedback for the paper.

### **Novelty of this work.**
The paper claims that previous works have not studied the implications of different LoRA ranks on the throughput, energy, and other system-level metrics. However, upon a quick search, several relevant works (see below) appear that are worth looking into before making this claim. These previous works bring into question the novelty of this paper.

- LaMDA: Large Model Fine-Tuning via Spectrally Decomposed Low-Dimensional Adaptation (EMNLP'24)
- ASPEN: High-Throughput LoRA Fine-Tuning of Large Language Models with a Single GPU (CoRR'23)
- Towards Green AI in Fine-tuning Large Language Models via Adaptive Backpropagation (ICLR'23)

### **Writing clarity**
I found it difficult to understand what the takeaways are from the study. For starters, before going into the methodology of the study, it would be helpful to know what research questions are going to be studied throughout the work. Moreover, while I appreciate the rich figures with several different metrics, I found it difficult to understand what I should take away from the figures, and unfortunately, the text surrounding the figures did not adequately distinguish the takeaways. It would be helpful to very clearly distill the findings from the figures (may bold, italicize, or have a small takeaways para, bullets, or box to make it clear).

There were several pieces of data referenced in the text that were not clear (or shown?) in the figures. For example, 64x reduction in storage requirements when going from rank 64 to 1. What is this storage requirement? Is it GPU memory? If so, the graph does not reflect a 64x reduction. If not, and instead it is disk storage, I don't think that should be the first piece of data that is written about (or at least, it should not be written about before the data shown in the figures). Moreover, the analysis refers to different batch sizes, but the figures don't show changing batch sizes. Are LoRA ranks and batch size the same thing? I don't think they are, but it is unclear from the writing.

### **Other notes**

1. After going through the figures, my understanding of the findings is the following: we should always choose lower rankings (between 64 - 256), since there doesn't appear to be a tradeoff between system-level metrics and model quality. If this is the case, is this not already known? Is there anything more to do with this finding, or is it simply to just use lower rankings for all fine-tuning jobs?

2. If fine-tuning is used for domain-specific tasks, is evaluating LoRA on a general Wiki-based dataset representative? Maybe for GPU-level metrics (e.g., throughput/power) this is fine, but for model quality, you state that you don't need large ranks to get good model quality. This may not be true for actual domain-specific tasks.

3. It would be helpful to explain what the limitations of the work are and future steps.

**Top Reasons To Accept The Paper:**

The paper takes a first step towards understanding the GPU system-level implications of different rank sizes for LoRA fine-tuning. The paper provides a good background on the theory behind LoRA fine-tuning. Moreover, the paper does a decent job of exploring the implications of rank on the compute and memory utilization, throughput, and energy of the GPU.

**Top Reasons To Reject The Paper:**

1. It is difficult to understand the takeaways from the 4 main figures presented in the paper. The text does not properly distill what the reader should takeaway with how LoRA rankings affect throughput, energy, and system-level metrics.
2. It is unclear why this work is novel. The authors claim the implications on system-level and GPU-level metrics have not been studied, however, with a quick search, several papers have studied this.
3. There are several references to data not shown in the figures but central to the study. For example, 64x reduction in storage requirements when going from ranking 64 to 1. I don't see this data in the figures.

---

### Official Review · Reviewer_CQFd · 2025-05-18
**LoRA Rank Sweep Impact on GPU metrics**

**Confidence:** 3
**Rating:** 5

**Detailed Feedback And Questions For Authors:**

The paper observes an interesting gap in the existing work. It attempts to provide an analysis for this. However, the evaluation and results are not very comprehensive. It can be improved further to include: LoRA adapting FFW layers too, include other models, include more fine-tuning tasks.

**Top Reasons To Accept The Paper:**

- Provides an evaluation of LoRA Rank to system level power/performance metrics along with quality benefits

**Top Reasons To Reject The Paper:**

- The evaluations are very limited and it is hard to convince that they generalize to other tasks or models.

---

### Official Review · Reviewer_W6Ls · 2025-05-19
**How Low Can LoRA Go. This work provides empirical experiments and analysis of LoRA fine tuning, especially focusing on varying LoRA rank sizes.**

**Confidence:** 3
**Rating:** 5

**Detailed Feedback And Questions For Authors:**

I like the effort on system performance analysis for fine-tuning tasks, where there are many parameters that need to be selected. I also think that this paper contains good useful practical tips in configuring fine-tuning experiments/scenarios.

The main experimental findings, as the authors described, may be that reducing the rank does not reduce quality too much while maintaining good system performance. But I was not very sure how academically strong the findings were. Also, just reading the paper, I was also curious how well this finding can be generalized for larger fine-tuning setups with multi GPUs.

Overall this paper is very well written. But I have few minor comments about the experiment plots.
- On the plots, some legend names were too general. It may be helpful to clearly define the legends on the plots and what metric was counted. For example, does “Compute” in the plot mean “gpu utilization”?
- The experiment plots are all relative values to the rank 2048 setting. But the text often describes raw numbers, and it was a little confusing to find the corresponding data points in the plots.
- Why the throughput/compute performance drop after the inflection point (batch size changes) was less prominent in SQuAD than that of Wikitext experiments? (I might have missed it though).

**Top Reasons To Accept The Paper:**

- Good empirical study of the system level performance of LoRA fine-tuning on an A100 GPU system, with well defined fine-tuning scenarios.

- Well-written paper. Writing and descriptions are very clear.

**Top Reasons To Reject The Paper:**

The specific experiments are well described. But, just reading the paper, I was wondering how generally the key experimental findings would be held on larger experiments. I was also not very sure if these findings are really new knowledge or academically strong.

---

### Official Review · Reviewer_2d36 · 2025-05-20
**Interesting sensitivity study for LoRA.**

**Confidence:** 3
**Rating:** 6

**Detailed Feedback And Questions For Authors:**

As pointed out in the future work, it'd be great to include more datasets. In addition, the author should also look into:
1. Multiple GPU finetuning for larger model. For example, the MLPerf benchmark uses SCROLLS GovReport and Llama 2 70B.
2. More HW options. Some end user might perform LoRA with their consumer devices (e.g., 4090), and others might use Hopper GPU, instead of Ampere GPUs. It'd be very interesting to see the trend of the best rank for LoRA changes across various platform.

**Top Reasons To Accept The Paper:**

LoRA is a practical and useful technique for model finetuning, and this paper provides the early performance study of LoRA to guide the ML practitioner to better use modern HW for LoRA.

**Top Reasons To Reject The Paper:**

Evaluation is still fairly limited in the breath of the dataset and system setup.